# Cooperation in the Conceptualization of Autonomous Strategic Initiatives: The Role of Managers' Intellectual and Social Capital

Emmanuel D. Adamides 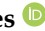

Management & Organization Studies, MEAD, University of Patras, 26504 Rion, Greece; adamides@upatras.gr

**Abstract:** The purpose of this paper is to explore how the social position of functional managers, as defined by their stocks of intellectual and social capital, influences their attitude towards cooperation for the integration of distributed knowledge in the conceptualization of bottom-up (autonomous) strategic initiatives. Bourdieu's social practice theory was employed for integrating the organizational conditions in the initiative conceptualization-as-knowledge-creation process. By developing and analyzing two case studies on strategic operations, it was found that the degree of engagement in productive cooperation, and hence the potential and effectiveness of functional managers as knowledge-creating agents promoting their particular interests, are influenced by their social position which in turn depends on the path of accumulation of their intellectual and social capital resources.

**Keywords:** strategic initiatives; knowledge-based view; practice perspective; social position; Bourdieu; operations managers

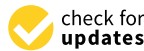



## 1. Introduction

In today's unpredictably fast-changing business world, strategic renewal through initiatives is a very frequent phenomenon [1,2]. Strategic initiatives are forms of corporate entrepreneurship that start with an idea and/or the recognition of an opportunity or threat and end with a form of approval [3]. Their development process aims at generating new knowledge, securing resources and acquiring legitimacy [4] to develop or renew resources and capabilities associated with competitive advantage [5,6]. Hence, they resemble but are different from projects that have well-defined start and end times and are more task-oriented and interdependent [7,8].

Initiatives may follow either top-down or bottom-up trajectories in organizational hierarchies [2]. Broadly, top-down (induced) strategic initiatives are directly associated with the specific processes by which strategy is realized, i.e., with what has to be done to implement a decided strategy, and they are frequently associated with disinvestment decisions. Bottom-up (autonomous) strategic initiatives are triggered by functional managers (operations, marketing, R&D, etc.) aiming at influencing the competitive strategy of a firm [4,9]. In early deliberations for agreeing on the scope of initiatives, functional managers interact with other initiative stakeholders to promote and coordinate their particular knowledge and interests [10] for their benefit and for the benefit of their organization [11–13]. As the objectives and interests of different functions are interlinked, frequently in contradictory ways [14], and knowledge is distributed, there is a need for cooperation in the definition of the exact scope and process of the initiative, i.e., its content.

The need for cooperation emerges in the distinct perspectives towards initiatives: knowledge-creating and conditioning/organizational views. The former is concerned with how initiatives are developed and knowledge is integrated [15], whereas the latter is concerned with how initiatives are selected [3]. According to the knowledge-creating perspective, initiatives are considered as knowledge particles that are modified as they move from the initial idea to approval. In the conditioning view, initiatives are considered constant and bounce back and forth between actors until they are selected for implementation.

So far, most of the research on initiatives in general, and bottom-up initiatives in particular, has undermined the conceptualization phase, instead considering their entire life cycle as a whole [6,16,17]. As a result, micro-level issues, such as the role of cooperation in initiative conceptualization as knowledge creation, how cooperation is achieved effectively and what the roles of the social position of initiatives' initiators and stakeholders are in this endeavor, have not been considered explicitly. The research objective and the aim of this paper is to fill this gap and explore how functional managers' social position in an organization, which is contingent on their knowledge structures, influences their attitude towards cooperation, on which the effectiveness of knowledge integration/creation and the initiative's legitimation depends. For the purpose of this specific research, we adopted the definition of productive cooperation as an exchange of knowledge and resources in which participants benefit from the encounter as much as the organization of which they are a part [18].

In this paper, we concentrate on a specific type of functional managers, namely operations managers, and we adopt a social practice perspective along the lines of Bourdieu's praxeology [19] to integrate organizational characteristics at the micro-level within the initiative conceptualization-as-knowledge-creation process. Hence, we assume that individual managers' stock of knowledge consists of *position*-related expectations, *dispositions* and *local knowledge* of particular (interactive) situations [10], created and modified in discursive practices carried out in the initiative development/conceptualization process, while applied to the knowledge content of the initiative through a set of socially embedded practices [20,21]. In a Bourdieusian context, the social position of a manager differs from their formal/institutional position in an organogram [22–26]. It is determined by their possession of additional intellectual and social resources, which have been accumulated throughout their extra-organizational life and differentiate them vis-à-vis others in similar positions [26–28]. These capital resources also determine the *orientation* of a manager in a position with respect to the organization's boundary (can be introspective or extrospective). Two organizational actors may be in the same position(s) but have different orientations. Following this, in this paper when not stated explicitly, an actor's social position will be assumed to include their orientation attribute.

To achieve the objectives of this research, we analyzed the development/conceptualization processes of two operations-initiated investment initiatives in two food processing SMEs in Greece. Since our objective was to investigate how managers in specific social positions cope in particular contexts, we used a case study research method [4,29]. SMEs were chosen as it was easier to isolate their initiative development processes and focus on specific managers for analyzing their micro-activities.

The original contribution of this paper is fourfold. First, regarding the initiative conceptualization-as-knowledge-creation-in-context literature in general, by adopting a Bourdieusian practice perspective, this paper shifts the level of analysis beyond "orthodox" meso-/macro-processualist and evolutionary approaches, which solely seek contingencies between managers' personalities and organizational characteristics [5–7,9,16], to the micro-level of individual managers' contexts and histories. Second, this paper provides a novel perspective on the knowledge-based view towards initiatives by abandoning dichotomous explicit–tacit knowledge [30–32], stressing the distributed nature of organizational knowledge held by different actors and its integration as a discursive practice of collaboration. Third, it contributes to the ongoing research on the role of managers' social positions, defined by the accumulation of intellectual and social capital, in the initiation, scope and support of change [24–27], by surfacing the influence of social position (and orientation) on willingness and commitment towards cooperation with other change agents to (co-)decide on the scope, and hence increase the effectiveness, of change. Finally, regarding a function's role in strategic change—operations, in our case—it contributes to understanding how autonomous initiatives influence competitive strategy. So far, although alignment as agreement on a change objective has been a central issue in the operations strategy

literature [12,33], with the exception of some very rare indirect references [7,13], it has not been considered from a strategic initiative's social practice perspective.

This paper proceeds as follows: we begin with a discussion of the role of functional managers in the strategic initiative development process. We continue with the presentation of Bourdieu's theoretical framework contextualized in the conceptualization process of functional strategic initiatives through the accumulation of knowledge and legitimation. Then, the methodology used is presented, while the section that follows depicts and analyzes two initiatives in two case study companies. The next section provides a comparative discussion of the two cases and develops an explanatory narrative for the influence of social position and orientation on productive cooperation in operations (functional, in general) initiatives' development processes. The last section outlines the conclusions of this research.

## 2. Functional Managers and Strategic Initiatives' Conceptualization Processes

Initiatives stem from individuals that seek to express their special skills, interests and ideas or advance their careers [9,16]. Functional managers of middle or higher rank depending on the size and structure of an organization, may want to act as strategic actors by emphasizing the importance of their function and their own stock of knowledge in the generation and deployment of initiatives. Operations managers as functional managers who are responsible for the management of an entire array of resources and processes [34], especially in SMEs [35], have many opportunities to do this; hence, they are frequently the initiators of bottom-up (autonomous) initiatives [2,13,36–38].

In order to overcome their usually assumed reactive role [33] and contest speculation, criticisms and resistance when they initiate change [11,18], operations managers need to cooperate with other initiative stakeholders in obtaining knowledge and resources. Moved by both dispositional (pre-reflexive action [39,40]) and purposive strategies [10,41], they use different means, such as rational justification, authority, politics and coalition making [16], to deploy their knowledge, influence the content and process of the initiative, enrich its initial goals, facilitate its deployment and increase the effectiveness of its outcome [3,6,17]. In this context, researchers have already noted that a manager's social position impacts the way and the effect of the means they use in the entire initiative life cycle to pursue change [26].

So far, initiative development has rarely been considered in vitro, conceptually disconnected from an initiative's entire life cycle [6,16]. There is a rich literature at the meso-level of analysis concerning "evolutionary" processes of initiatives' selection and "survival", principally for top-down initiatives [5]. They assume given and relatively constant initiative content as the knowledge particle, a static selection environment and impersonal and ahistorical strategy (selection) processes. In addition, they place personal interest-driven and disposition-conditioned influences and struggles, as well as knowledge and power distribution processes, organizational politics and the wider sociopolitical environment in the background [5,16]. Research at this level of analysis seeks contingencies that stress the importance of internal and external organizational conditions [3,9] without opening the "black box" of wider managerial decision-making activity, as knowledge-creating/integrating activity, to provide explanations for this activity, i.e., how the proponent(s) of initiatives tries to sell it vertically and horizontally in the organization, and how stakeholders of the initiative are engaged in the buy-in of the initiative by modifying it.

In fact, an initiative is constituted of an idea to do something about an issue/problem and a set of actions/activities towards realizing this idea. In the development process, the proponent(s) and supporter(s) of the initiative need to cooperate to acquire resources (capital and other assets, such as content knowledge, knowledgeable managers and employees) and create knowledge (e.g., to enrich the idea and provide more alternatives to choose from, as well as criteria for selection) that enrich its goals, facilitate its deployment and increase the effectiveness of its outcome [3]. The initial idea is a knowledge particle undergoing modifications and the outcome of the initiative development process is proposals about

interventions in organizational structures and processes, as well as the acquisition or development of novel resources and capabilities that have the support of other stakeholders beyond the initial proponents.

In this context, so far, managers' social position has been examined either with respect to the initiation moment of the initiatives or in the framework of their entire conceptualization–implementation cycle [23,25,26]. The practice perspective adopted in this paper examines the initiative development/conceptualization process explicitly and considers practices in content development, argumentation, presentation, reporting, conflict resolution, etc., as expressions of pre-reflexive strategic activity complemented by rational strategic choices in interactive situations. Such a perspective emphasizes the roles of the social position, dispositions and interests of the participants as determinants of converging deliberations and decision making regarding the scope of the initiative [26].

For the sake of analysis, the initiative conceptualization process can be considered as a sequence of stages [36,42] resembling the classical (rational) decision-making process [43]. Hence, it comprises the following stages: problem identification (stage A); speculation on and amendments to the initial proposal (idea) and development of alternatives (stage B); discussion and argumentation of the selection criteria of the alternatives, before the (initial form of the) initiative (the product of knowledge integration) is selected (stage C); and eventually, approval and implementation (stage D).

## 3. Bourdieu's Practice Theory: Social Position as a Generator of Discursive Practices

For Bourdieu, a firm is not a homogeneous entity that can be treated as a rational subject [28,44], nor do all its members have the same knowledge stock [45]. Its strategy is guided by the outcome of the conflictual relations among the different organizational actors who have different interests, stemming from their social background, and who are bound to the interests of the organizational functions/units to which they belong or manage [44]. Actors are engaged in practices (habitual activities) in fields of practice (the organization or some organizational unit), whose structure they either want to maintain or change. Practices are carried out according to the capital they own (positional knowledge) and their habitus or envelope of dispositions for action. Dispositions depend on capital and drive further capital accumulation [28]. The field, capital and habitus form a system and practice cannot be understood without reference to all three and their interrelationships. Actors—managers and employers—belong to many different fields at the same time, the relative importance of each field being determined by its position in the field of power.

The social position of an actor in a field is determined by the form and amount of capital that they hold. Capital may have many forms (economic—financial resources that agents can mobilize; bureaucratic—associated with the possession of formal position(s); social—involvement in networks; technical—knowledge and skills related to technologies; organizational—knowledge of procedures and rules; informational—privileged access to knowledge [46] and different origins (internal or external to the field in which it is deployed)). Symbolic capital is the combination of the forms of capital whose specific meaning, volume and composition indicate what counts more in a specific field, at a particular instance of time. The habitus, on the other hand, is what actors want to do without clear rationale in order to take more favorable positions in the field by accumulating the form of capital that their habitus dictates. The habitus, as an expression of the past and a trajectory, provides a sense of the "space of possibilities open" to someone in a given field of practice. It can be distinguished into "primary" (accumulated though one's lifetime) and "specific" (accumulated through specific organizational settings/fields) [28]. Through the habitus, Bourdieu explains what makes agents want change. However, only power/capital drives change. Other organizational scholars extended Bourdieu's framework of knowledge-creating activities by arguing that habitus-based pre-reflexive strategic behavior needs to be complemented by rationale strategizing [10,41].

On the basis of the above discussion, the strategic behavior of managers in initiative deliberations can be understood as the outcome of three constituent forces and their

corresponding forms of knowledge: (1) behavior stemming from their normative position/role in the organization/field and strategic interested actions/practices (interested strategizing) [40] for changing (or maintaining) the structure of the formal organizational field, (i.e., taking a better, more powerful, position in the organization's strategy discourse) [39,47]; (2) dispositional strategizing (or position taking) stemming from the social position of an individual manager as a result of their capital configuration formed throughout their tenure in the organization, as well as in other social spaces, in parallel or before entering the field of the particular organization, i.e., their broader culture [28]; and (3) rational–interactive strategizing employed in specific interactive situations (episodes), in the framework of the above two strategizing processes. In the context of this paper, the institutional/formal position of operations managers and their dispositions towards the dominant structure of the field are what triggers their interest in change through initiatives (*why* change), rational strategizing is involved when the choice of the timing to start the initiative is made and when the form and the content of possible alliance(s) are decided (*when* change), whereas dispositional strategizing (position taking), as a result of the forms of capital that a manager as an individual carries, is responsible for their behavior in the process (e.g., cooperative or not) of the development of the initiative (*how* change is promoted). Understanding behavior in change initiative development requires understanding all three dimensions.

By the same logic, the social position of a manager in an organizational field is defined by the combination of the forms of capital that are engaged in the three elements of strategic activity: (i) symbolic capital, usually in bureaucratic or economic form, that comes from and defines the formal position of the function and its manager in the organization; (ii) the configuration of other forms of capital (can be of any form) endowed within and outside the organization; (iii) (principally) social capital for mobilizing networks of individuals to form alliances in interactive–rational strategic situations. These forms of capital are determinants of the social position and orientation of a manager, which through the habitus, influence decisions and actions concerning not only if and when to initiate change, but also cooperative practices in the entire initiative development/conceptualization process. Put in another perspective, there is a role for formal and informal positions in knowledge integration. This forms the basic proposition explored in the two case studies depicted below.

## 4. Methodological Approach

### 4.1. Research Design and Process

We used a two-case research process [29], consistent with the practice perspective in organizational phenomena [48]. Field research was carried out in two manufacturing SMEs in Greece. The two companies were selected from an initial sample of twenty companies planning process technology investments/disinvestments in 2012 in the context of the unfolding economic crisis. Both had very similar business activity (sea food processing) allowing for intrasectoral comparison. It is important to note that their operations managers left the companies in 2016, a fact that allowed them to comment on events ex post at a later stage more objectively. At that time, they employed 100 and 108 people, respectively.

Initially, we examined and recorded four production–technology investment initiatives (two in each company) initiated by operations, whose conceptualization phase lasted from one to four months. This helped in building the general context of initiative development. Then, two of these initiatives, one in each company, were chosen by the operations managers involved as being more representative of what was common practice. Table 1 below summarizes the research effort (observational visits and interviews) put forth for the development of the two cases. Interviews were semi-structured and lasted between one and two and a half hours each. Upon request by the two companies, interviews were not recorded and not fully transcribed. However, detailed notes were taken and at the end of the day, or the next day, key observations and phrases were shared in a structured reflexive way. A number of documents were also consulted on an

ad hoc basis when discussion led to them. Notes were taken. Commented observations and interviews were analyzed ex post.

**Table 1.** Summary of research activity.

|  | **Visits** | **Days per Visit** | **Telephone Interviews** | **Total No. of Interviews** |
|---|---|---|---|---|
| FOOD1 | 4 | 1 | 6 | 8 (two of which were *ex post*) |
| FOOD2 | 4 | 1 | 7 | 5 (two of which were *ex post*) |

### 4.2. Data and Codes

The first stage of analysis involved coding for the social position and orientation of the managers involved in the initiatives' development. Positions were associated with specific capital structures. Hence, the combinations of the forms of capital which generated individual practices had to be considered explicitly. Codes for positions were developed manually from reading the notes carefully and keeping in mind Bourdieu's forms of capital discussed in Section 3 [24]. Certain themes that were identified in the notes were consolidated before being assigned to a specific form of capital. The codes developed for the various forms of capital are given in Table 2.

**Table 2.** Data structure.

| First Order Codes | Theoretical Categories | Aggregate Theoretical Dimension |
|---|---|---|
| Involvement in company's investment decisions<br>Involvement in company's budget development | Economic capital | Position |
| Diversity and orientation of intraprofessional and intrasectional relationships | Social capital | Position |
| Diversity and orientation of relationships spanning interprofessional and intersectional boundaries | Social capital | Position |
| Knowledge of formal and informal rules in organization (institutional) | Organizational capital | Position |
| Knowledge of formal and informal rules in doing business (in sector) | Organizational capital | Position |
| Formal (institutional) position | Bureaucratic capital | Position |
| Status in society | Bureaucratic capital | Position |
| Experience in holding positions related to formal education/training | Technical capital | Position |
| Formal education/training in relation to position | Technical capital | Position |
| Privileged access to internal company information (financial, industrial relations, strategic, etc.) | Informational capital | Position |
| Privileged access to market information | Informational capital | Position |
| Perception of importance of function and company<br>Expressed loyalty to functional staff<br>Argumentation based on loyalty and tenure in function | Belonging | Disposition |
| Expressed support for the needs of fellow managers<br>Affinity towards participative decision making<br>Tendency carefully listen to opinions of fellow managers<br>Expressed awareness of common future in company with fellow managers<br>Tendency to seek feedback from managers and staff | Collegial | Disposition |
| Reference to formal company rules in intracompany social encounters<br>Expressed reluctance to be engaged in informal activities outside their area<br>Use of professional standards and repeated reference to them | Professional | Disposition |
| Decision making and actions based on available data for the situation<br>Refusal of others' arguments when based on intuition<br>Tendency to inquiry<br>Use of checklists, tables, etc. | Scientific | Disposition |
| Expressed preference for written communication<br>Expressed tendency to develop procedures<br>Construction and use of time schedules | Administrative | Disposition |
| Use of project control tools<br>Frequent inspections and checks | Controlling | Disposition |

**Table 2.** *Cont.*

| First Order Codes | Theoretical Categories | Aggregate Theoretical Dimension |
|---|---|---|
| Early notification of initiative proposal<br>Explicit consideration of implications to other functions in the initiative proposal<br>Signaling for willingness to discuss alternatives and modify initiative<br>Use of language and form understandable by other stakeholders | Productive cooperation in initiative proposal (stage A) | Practice |
| Proposing alternatives/modifications with explicit reference to initial proposal<br>Introduction of alternatives/modifications in the same language as original so that can be understood by all<br>Proposing and arguing for selection criteria in a form compatible with a common argumentation scheme<br>Proposing and arguing for selection criteria in a language that can be understood by all<br>Proposing quantifiable selection criteria that have been tested before | Productive cooperation in the proposal of alternatives (stage B) | Practice |
| Keeping a constant argumentation scheme for a preferred selection<br>Listening to alternative criteria and preferences<br>Proposing criteria that concern other functions and the organization as a whole<br>Not using status/position to impose criteria and select alternatives<br>Seeking consensus on criteria and/or alternatives<br>Democratic conflict resolution by democracy | Productive cooperation in the proposal of criteria and selection of alternatives or modifications (stage C) | Practice |
| Avoiding playing the blame game if dissatisfaction is expressed by top management<br>Engaging other stakeholders in responding to speculation and inquiries by top management<br>Supporting collaboratively developed initiatives | Productive cooperation in initiative approval (stage D) | Practice |

After coding for the different forms of capital, we searched the literature for the dispositions of both middle managers and functional managers. Taking into account the descriptions and analysis of the role and tasks of operations/manufacturing managers [13,33,34,49,50] and middle/functional managers in general [36,38,51–53] as presented in the literature, the dispositions of the habitus of the four managers in the two companies (operations and marketing managers as main stakeholders of the initiatives) were inductively distilled into six broad types: *belonging*, *collegial*, *professional*, *scientific*, *administrative* and *controlling* (Table 2). A belonging disposition means that the manager has a sense of loyalty or sentimental attachment to [54], and a psychological contract [18] with, the function that they are managing and their primary concern is to promote its interests. In many cases, this sense of loyalty extents to the entire organization. A collegial disposition implies a concern for colleagues' needs, for contribution to boost team synergy, for listening and using feedback provided by colleagues and for promoting a friendly, non-confrontational climate. The professional disposition implies that managers stick to the rules of their supposed role in the organization. They do nothing more and nothing less, respecting their role's attributes, independent of whether they agree or not with all of them. They do not like to be involved in "others' business". Managers having the scientific disposition see their role in connection with their knowledge base and the training they have received. They frequently refer to management theories, they like numbers and are difficultly persuaded by intuition and vague arguments. Administrative disposition characterizes managers that see administrative work (filling forms, checking lists, producing guidelines, reports, etc.) as a legitimation of their activity. Managers having this disposition, as a result of the accumulation of organizational capital, know rules and take advantage of bureaucratic procedures. Finally, managers having a controlling disposition view their work as a means to create order. Frequently, this disposition emerges as a reaction in organizational environments where roles and work processes are quite flexible.

Argumentation practices as knowledge-creating and integrating practices [55–57] in initiative support and conflict resolution were characterized according to the justification claims (schemes) employed, i.e., expert opinion (accept claim because someone is an expert), popular opinion (accept because it is generally accepted as true), analogy (accept because it works in a similar situation) and causal associations (A works because B works, and there is a positive correlation between the two) [58]. As they are part of the position-taking structure (culture) of an organization, they were positioned in relation to the dominant argumentation scheme of the organization (the scheme used to justify arguments more frequently and legitimize opinions and actions [59]).

Finally, on the basis of the literature on effective teamwork and cooperation, a set of productive cooperation practices along the dimensions of trust, healthy conflicts, commitment, accountability and results orientation [60,61] were defined as depicted in the lower part of Table 2. These include signaling intentions, showing willingness to discuss alternatives, suggesting proposal(s) in forms easily understandable by other managers, introducing alternative proposals with explicit reference to the original idea, keeping a constant argumentation scheme, democratic conflict resolution and avoiding playing the blame game.

## 5. Short Case Descriptions

### 5.1. Case A—FOOD1

#### 5.1.1. The Firm and Its Environment

FOOD1 was established 35 years ago. At the time of the study, it was processing imported fish as well as fish from domestic fisheries. The initiative concerned an investment in a new line for the production of ready-to-consume fish dishes (after closing down its shellfish processing line). The technological level of the company was rather low. Prior to the initiative, FOOD1's operations' strategic priorities were a high quality and responsiveness/speed. The strategic objective behind the initiative was to diversify through new product lines and increase flexibility. FOOD1's dominant argumentation scheme was "analogy".

The official title of the operations unit's manager was Production and Distribution Manager (PDM), signifying the importance given to the challenges of distribution in small quantities to local shops. The PDM was 55 years old, held a food technology degree from a British university and had joined the company 25 years ago. He had held different positions in the company, including those of being a purchase manager and sales manager and had a tendency to compare domestic business practices with those abroad.

FOOD1's marketing manager had a degree in economics from a domestic university. He was 43 years old and an incumbent of the company in a position that never had a high status and power. The company relied heavily on external services for its marketing, and this was a situation which the marketing manager wanted to change. His relations with other managers and employees were informal and cooperative. He frequently referred to the practices of other companies of the same size and structure.

#### 5.1.2. The Initiative Development Process

A stage-by-stage description of the initiative development process is given below. For reasons of space, in Table 3, illustrative evidence is depicted only for the operations manager. For both cases, in Tables 3 and 4, superscript denotes that the data: (1) are the result of observation; (2) come from interviews with the specific manager; (3) come from interview(s) with (an)other manager(s); and (4) were gathered from other company sources (e.g., documents).

*Stage A*—The proposal by the production and distribution manager (PDM) (code OM1) was for a moderate investment in a new flexible production line for ready-to-consume fish starter dishes. He provided capacity estimations and brief specifications and argued for endogenous financing through the capital obtained from selling the shellfish lines. OM1 indicated that the technical capability for operating the new line was already available

and made clear that the new line would be compatible technologically with the rest of the company's infrastructure.

**Table 3.** Illustrative evidence for OM1.

| First-Order Codes | Theoretical Categories |
|---|---|
| He had held different positions in the company including those of being a purchase manager and sales manager. [4] | Social capital—external |
| "Although he was in operations, he knew very well the procedures for exports." [3] | Organizational capital—external |
| "I had mastered purchasing. I could find the best supplier of everything." [2] | |
| He held a food technology degree from a British university. [4] | Technical capital—external |
| He was involved in the local society. [4] | Informational capital—external |
| "He was professional. Good at his work, I didn't count on him for support beyond the departmental lines." [3] | Professional disposition |
| "I was never interested in the company's internal politics. I just did my work in the best possible manner." [2] | |
| "He studied marketing. He didn't know anything about automated handling systems." [2] | Scientific disposition |
| "He always developed checklists of the documents required for the applications in national and regional authorities and placed them in the cover of dossiers where he kept the documents." [3] | Administrative disposition |
| "After the late developments my interest was to keep the company united as a family to get over the effects of the crisis" [2] | |
| | Collegial disposition |
| The operations manager and the marketing manager had a close relationship stemming from the long tenure they both had in the company. [4] | Social capital (figurational structure)—internal |
| A file containing vendor brochures, clips from international industry magazines, as well as rough cost–benefit calculations was sent to stakeholders. [3] | Practice—Productive cooperation in initiative proposal (stage A) |
| Argumentation was based on analogy. [1,4] | |
| He provided more analytical data in three scenarios (with corresponding rough probability estimations): pessimistic, optimistic and most probable. [1,2,3] | Practice—Productive cooperation in the proposal of alternatives (stage B) |
| Argumentation was based on analogy. [1,4] | |
| Voting for conflict resolution. [1,2,3,4] | Practice—Productive cooperation in the proposal of criteria and in selection of alternatives or modifications (stage C) |
| Collaboration—changed initial proposal. [2,4] | |
| Not much involvement in initiative approval. [3] | Practice—Productive cooperation in initiative approval (stage D) |

**Table 4.** Illustrative evidence for OM2.

| First Order Codes | Theoretical Categories |
|---|---|
| "He has mastered the visible and the hidden processes of FOOD2 with his unit at the center." [3] | Organizational capital—internal |
| He had always been in the production department (15 years) and acquired technical knowledge by practice. [4] | Technical capital—internal |
| "I was the oldest incumbent in the meeting and I was not asked to express my view about something that concerned the future of this company." [2] | Belonging disposition |
| "I spent my entire life in this company and if nothing exceptional happens I will retire from this company." [2] | |
| "When I came to the company, the factory was in a chaos. Nobody knew what was responsible for, nobody knew how much the factory produced every day; nobody knew how many units were missing in the orders received. I had to put everything under control and that's what I do since then. This is what I leant in the army!" [2] | Controlling disposition |
| "I proposed to write a report to the MD on the different alternatives that we considered, explaining why we had to reject them, one after the other." [2] | Administrative disposition |

**Table 4.** *Cont.*

| First Order Codes | Theoretical Categories |
|---|---|
| Initial proposal in electronic form sent via e-mail. [1,2]<br>The initiative proposal was a surprise. [2]<br>The mail sent included two contractors' offers and preliminary cost analysis for building the cold rooms. [2,3]<br>Argumentation was based on expert opinion. [1,2,3] | Practice—**No** productive cooperation in initiative proposal (stage A) |
| There was no accommodation of views on alternatives. [1,2,3] | Practice—**No** productive cooperation in the proposal of alternatives (stage B) |
| Issue changed to risk management. [1,2,3] | Practice—**No** productive cooperation in the proposal of criteria and in selection of alternatives or modifications (stage C) |
| The OM presented his own version of the story from its conception until the initiative was finalized. [1,2] | Practice—**No** productive cooperation in initiative approval (stage D) |

A file containing vendors' brochures, clips from international industry magazines, as well as rough cost–benefit calculations was used for promoting the initiative. A brief memo accompanied the file sent to the managing director (MD), the finance manager (FM) and the marketing manager, who were already vaguely aware of the intentions of the PDM. The main argument of his proposal was that "successful companies of the same sector abroad made this move with the same technologies" (analogy scheme).

*Stage B*—Alternatives in the form of smaller scale investments with lower flexibility were proposed by the finance manager (FM), who expressed his concerns about the cost and time of potential returns of the investment in a memo distributed to all company managers and supervisors. The PDM responded by providing more analytical data, but the FM insisted that the finances of the company needed capital as a safety net during the crisis. His main argument was that "this was what other domestic companies do" (analogy scheme).

In a strategic meeting where the initiative was discussed, the marketing manager (code MM1) intervened arguing that the existing product lines had limited potential for both the domestic market and for exports (argumentation scheme of expert opinion and analogy). However, he also expressed some concerns about the real costs and returns of the investment.

*Stage C*—As it became obvious that medium-term costs and returns were the criteria for selecting the most appropriate "size" of investment, the OM1 and the MM1 agreed to work together on the development and assessment of scenarios to prove that, if nothing was done, the shrinking markets of the existing product lines would gradually absorb the capital available from the sale of the old line. They did so, and they came up with a fully developed proposal for a smaller but scalable line which could be partially subsided through EU regional development funds. After that, the FM agreed to forward the proposal to the managing director for approval.

*Stage D*—After short consideration and a couple of telephone calls to the proponents of the initiative for clarifications, the managing director agreed to move forward with the investment.

*5.2. Case B—FOOD2*

5.2.1. The Firm and Its Environment

The company had a very long history in the processed fish industry, especially in the production of canned cured fish. It paid particular attention to the production technology used and to the adoption of international quality standards and accreditations/certifications. Over the last years, the strategy of the company was towards extending its range of products. FOOD2 had a tradition of flexible work arrangements and the operations strategy of the company before the initiative was towards quality. The initiative concerned investments in automation and refrigeration (cold rooms) to reduce costs through volume production

(produce-to-stock). The company was doing well in Germany and managed to secure financing for the investment through the intervention of its distributor there. Its dominant argumentation scheme was "expert opinion".

The operations manager had been an incumbent of the company for about 15 years. He was 40 years old and had no formal qualifications for the position held. During his tenure in the company, he had always been in the production department and gradually promoted from supervisor to production manager. The operation (production) function was considered as being of low to medium importance for the company.

The marketing manager was 44 years old, held a degree in marketing and had almost ten years of experience in the food industry before coming to FOOD2. In the company, he held the position of marketing manager for 3 years.

5.2.2. The Initiative Development Process

A stage-by-stage description of the initiative development process is given below. Again, for reasons of space, illustrative evidence for the operations manager (OM) only is depicted in Table 4.

*Stage A*—The initiative of the OM (code OM2) concerned investment in cold storage rooms and automation of the smoked fish production lines. Assuming the role of an "expert", the argumentation put forward by OM2 was that these investments would reduce the unit cost of basic ingredients and increase competitiveness as the quality of products was already taken for granted.

An e-mail message was sent to the marketing manager (code MM2) and market development manager (MDM) with sales figures abroad as attachments. They showed an increasing dynamic as well as two contractors' offers and a preliminary cost analysis for building the cold rooms. Two weeks later, OM2 sent a document that he had developed with the help of the chief accountant, listing product unit cost calculations based on scenarios of production at a larger scale.

*Stage B*—After receiving OM2's proposal almost by surprise, MM2 responded that he was already concerned with an excessive production capacity, and he included much lower sales in his new marketing plan, due to shrinking markets in the global economy in a recession. He also made notice of the risks involved in maintaining large inventories. For him, the actual capacity required for cold rooms was much lower, and automation was just an option for the future. His argumentation was based on his own data and analysis as a marketing expert (expert opinion argumentation scheme). In the following five weeks, there were eight face-to-face discussions and twelve e-mail exchanges for the support of the two alternatives, but no real accommodation of perspectives was achieved. The OM insisted in his maximalistic behavior and no attempt for accommodation was made by discussing lower capacities than those proposed.

*Stage C*—The inconclusive exchanges led both men to bring the issue indirectly to a wider audience. The initial exchanges of views and arguments were followed by two strategic group meetings, in which the discussions were initially focused on capacity levels and costs. Gradually however, when other managers got involved and expressed their concerns too, OM2 shifted the issue to risk management. He argued that automation and inventories are risk reduction moves. Based on this criterion and after the OM was unable to support any risk-related scenario for the future with reliable information, he withdrew and a decision was taken for smaller-capacity cold rooms. This stage lasted almost three weeks.

*Stage D*—After OM2 presented the initiative to the MD, complemented with his own rejected proposal, the MD requested the views of MM2 and the other managers involved, expressed in person. After this, the final form of the initiative was approved by the MD.

## 6. Cross-Case Analysis—Main Findings and Explanations

Based on summarizing the two cases in both organizational fields, economic capital formed the dominant form of capital linked to power. It was concentrated in the hands of the managing directors and a handful of top executives. The undertakings of the initiatives by the two operations managers were triggered by their low position in the field and their corresponding dispositions [40], as well as by strategic calculation regarding their timing. They wanted to change the structure and power distribution of the organizational field and improve their position in the fields. The initiative triggering event was for OM1, the sale of the old production, whereas for OM2, the availability of financing.

In both cases, entering the deliberations-led knowledge integration process, operations managers had stakes in and ought to have cooperated in achieving strategic alignment with the marketing managers who, in principle, were not opposing the initiatives. In both companies, operations were, more or less, in the same hierarchical position: low in FOOD1 and in a slightly better position in FOOD2. Both marketing managers had social positions and a habitus leaning towards cooperation (Table 5). They wanted to modify the initiatives in a favorable way as much as possible for them without, however, cancelling change. So, overall, it was up to the operations managers to lead the initiative development process in the direction they wanted.

**Table 5.** Cross-case comparison.

| Theoretical Dimension | OM-FOOD1 (OM1) | MM-FOOD1 (MM1) | OM-FOOD2 (OM2) | MM-FOOD2 (MM2) |
|---|---|---|---|---|
| **Positions** | | | | |
| Economic capital | Low | Low | Low to medium | Medium |
| Cultural capital | Social (external) Organizational (external) Technical (external) Informational (external) | Social (external) | Social (internal) Organizational (internal) Technical (internal), Bureaucratic (internal) | Social (internal and external) Technical (external) Bureaucratic (internal) |
| **Dispositions** | Scientific Professional Collegial Administrative | Collegial Professional | Belonging Administrative Controlling | Scientific Belonging Collegial |
| **Figurational structures** | | | | |
| Interactive context/strategizing | Strategic engagement of Marketing Manager | Active involvement in initiatives | Local community contacts | Status in Marketing community |
| **Practices** | Argumentation was based on analogy Use of traditional means of communication Use of data in arguments Support conflict resolution by democratic means (voting) | Argumentation based on expert opinion and analogy Use of data Support conflict resolution by democratic means (voting) | Argumentation based on expert opinion Ambush tactics Conflict resolution by changing issue | Argumentation based on expert opinion Support conflict resolution by power (reference to the position of Marketing in FOOD2) Systems thinking |
| **Dominant argumentation scheme** | | Analogy | | Expert opinion |

### 6.1. The Importance of Sources of Capital in Cooperation for Change

In leading the process of conceptualization of the initiative (agreeing on the scope of change), the two functional managers exhibited different behaviors: OM1 was cooperative while OM2 was distant and less cooperative. These divergent behaviors can be explained either as resulting from dispositions towards cooperation, activated by specific environmental conditions [62,63] or as the result of consistently practicing cooperation in everyday life [61]. In the relational Bourdieusian praxeology, the two perspectives are interlinked in a dialectic process. Dispositions lead to practice, and practice leads to dispositions (the habitus) though capital formation [28]. Hence, managers' dispositions produce practices of cooperation in the conceptualization of autonomous initiatives which are influenced by the social-position-defining capital resources, which, in turn, have been formed through practice in various fields of practice.

As was already indicated, cooperation is an exchange of knowledge and resources which is beneficial for those involved, as well as for their organization(s). In situations such as the ones depicted above, where those engaged in a common activity have different perspectives, knowledge stocks and conflicting interests, when cooperation prevails, commodities of power *are not* directly exercised for the resolution of the situation and synthesis is sought—it is not beneficial for the organization in the long run [64]. Cooperation necessitates a willingness and ability to understand the situation in terms of the other. Common understanding, in turn, is facilitated by the objectification of situated meaning [61], which, in turn, implies the availability of *pluralistic* (constituted by different elements/knowledge particles) and *transposable* intellectual resources from those involved in the situation.

*Transposability* across fields is the most important property of capital, and capital has the highest value when it is transposable, i.e., it can generate practice across different fields [65]. Transposable capital is not exclusive to specific fields, thus resulting in a weak embeddedness of its holder in a particular field. It is also associated with complex cognitive structures that help actors to make sense of and respond to a variety of situations in different fields of practice [20,66]. The opposite is true for capital formed in a single field or in a small number of fields: it is field-specific, homogeneous and has low transposability. It is associated with a high *centrality* of cognitive schemes, leading to strong embeddedness in specific fields and good responsiveness in known, stable situations [66]. Clearly, it is expected that social positions defined by this sort of capital lead to a habitus towards reluctance to cooperate in the *change* situations described (their scope of) in different, unfamiliar terms from those that of which their holders can easily make sense.

### 6.2. Social Position and Dispositions for Cooperation in Change Initiatives

OM1's social position was defined by capital primarily formed externally to FOOD1. His technological capital was initially accumulated through his formal training in a British university and he spent much of his tenure in the company in outward-oriented positions (as a purchase manager and sales manager) at the boundary of the firm, "facing" suppliers and customers. This resulted in the accumulation of social and organizational capital with an extrospective flavor and in shallow embedment in the organization. By following developments abroad, he also accumulated informational capital, again with an outward perspective. The accumulation of technical capital through formal training contributed to his scientific disposition. As he held positions at the boundary of the firm, and having to deal with many managers and external stakeholders in formal and informal processes (thus accumulating organizational capital), he had developed a professional disposition with administrative attributes which allowed him to manage the complexity of the different contexts which he had to face. In addition, having worked with different people, he had developed an appreciation for the importance of the human factor of organizations, thus also developing a collegial disposition.

On the other hand, OM2 had a social position defined by capital resources developed internally. He had accumulated his technical capital through his experience in the production function of FOOD2 and had not received formal training. So, his expertise and capabilities were, to a large degree, specific and embedded to FOOD2's processes. The source of organizational capital was internal, as he mastered formal and informal rules and processes in the immediate neighborhood of his unit. Relative to OM1, his bureaucratic capital was higher as the production department was considered somehow important for the company. His internally accumulated capital resources resulted in a belonging disposition because he thought of himself as a company institution and could not imagine himself working in a different organization. In addition, as he could not easily comprehend complex situations due to his lack of formal training, extent of experience and narrow perspective, he had dispositions towards an administrative logic and control.

Given the appropriate conditions in a field, collegial, professional and scientific dispositions can easily be associated with an inclination (habitus) towards cooperation, whereas belonging, administrative and controlling dispositions can be associated with reservations

towards cooperation. A collegial disposition assumes the values of trust and teamwork, a professional disposition implies respect and recognition of the other's space and time, while a scientific one assumes a common objective ground for recognizing and accepting truth. On the other hand, a belonging disposition puts the interests of the group or organizational unit first and against the interests of others, an administrative disposition is associated with reluctance to act beyond formality, something that is frequently required in cooperative conflict resolution, whereas controlling is associated with a reluctance to give space and control to others.

Hence, based on the description of the initiative development process and the evidence depicted in Table 3, the social position of OM1 articulated a habitus towards cooperation, expressed in cooperative practices. Before making his intensions public, he first informally discussed the possibility of launching an initiative and then communicated a fully documented proposal. He kept a constant argumentation scheme in line with the dominant one of the firm and encouraged, used data/facts to support his claims and participated in a democratic process of conflict resolution (voting). In order to promote the initiative, OM1 strategically invoked his personal relationship with MM1 (social capital) when opposition was expressed by the FM. On the other hand, as depicted in Table 4, OM2 used a sort of "ambush strategy" in launching his initiative and tried to bring other managers in on a situation of *fait accompli* (secret preliminary discussions with and offers of equipment vendors). He assumed the role of an expert in argumentation without, however, providing clear justifications. He changed the issue into risk management and tried to present to the MD his own version of what had taken place in the initiative development process. In general, his behavior was characterized by selfishness and mistrust, and his belonging disposition was confined to the unit he was managing. Overall, in contrast to OM1, OM2, by not being cooperative, was unable to take advantage of his position and effectively place his organizational unit in an active role, as far as strategic change was concerned.

Table 5 summarizes the points depicted above and provides a cross-case comparison based on the pairs of managers that were involved in each case.

## 7. Implications for Practice

The cases presented and the analysis that followed suggest that senior managers must pay close attention to the social position of (functional) managers who are engaged in change initiatives. Their social positions, as defined by their stocks in intellectual and social capital, will be responsible for their dispositions towards productive cooperation, enabling knowledge integration from different sources, i.e., extending the scope and effectiveness of change.

Our research suggests that managers with professional and living experience in diverse settings, as well as those that have encounters with individuals of different ethnic and processional cultures, are more likely to cooperate and are more willing to share/integrate their knowledge in change initiatives because of the transposability of their capital stocks. To increase the transposability of the capital stocks of their operations managers, senior managers need to encourage their engagement in different tasks and as much as possible, at the boundary of the organization. This means that operations managers must be freed from their reactive and passive role and be engaged in tasks of creativity and continuing dialogue with many different actors (e.g., marketing managers), including external ones. This will renew capital resources and encourage dispositions towards cooperation and knowledge integration.

## 8. Conclusions

In this paper, we theorized that the potential and the effectiveness of functional managers as change agents who can influence the direction of change though initiatives are influenced by their social position and associated knowledge stock, in general, and their orientation, in particular. We considered initiative conceptualization as a set of discursive practices of collaboration for distributed knowledge integration. We also considered so-

cial position as the result of multidimensional capital accumulation, and orientation as dependent on the path of capital formation. Cultural/intellectual capital as the stock of knowledge of diverse forms, formed by their exposure to diverse fields external to the specific organization to which they belong, is characterized by plurality and transposability, leading to weak embeddedness in an organization's operational logic and processes. It also leads to openness towards alternative contributions of knowledge regarding the scope and process of change, to a disposition for cooperation and to the employment of associated interactive strategizing practices. This enables functional managers to have a more active role in strategy making, independent of their position in the organogram. On the other hand, managers with internal organizational paths of capital formation, lacking exposure to a diverse range of fields of practice, are in social positions with an introspective orientation. Introspective orientation results in strong organizational embeddedness through mastering a limited number of concepts and processes specific to the particular organization. Alternative ideas/proposals are considered distant and viewed with suspicion, leading to a reluctance towards cooperation with others who have different views and interests in change initiatives.

To arrive at these conclusions, we used Bourdieu's praxeology as an integrating framework of the knowledge-based and conditioning views of initiatives and as an analytical lens in two case studies of initiatives led by operations managers in SMEs. Clearly, for the extensive generalization of these results, further empirical research in a larger sample with different functional managers in different contexts is required.

## 9. Further Research

In this paper, we explored how the social position of functional managers, defined by their stocks of intellectual and social capital, shapes their attitudes towards cooperation and knowledge integration in the conceptualization of bottom-up strategic initiatives. Hence, this paper opens a new direction in the analysis of strategic initiatives, beyond the conditioning and "pure" knowledge-based views. To this end, further research is required to better understand the rational–interactive strategizing of managers' behavior, which is contingent on very specific situations. In addition, given that what is presented in this paper is based on explorative case studies, further empirical research is required to concretize the obtained results [67]. Moreover, the role of technology, in general, and ICT, in particular, has to be considered explicitly. That is, it is imperative to explore how technology, such as AI and data and knowledge bases, can be integrated in the human-driven knowledge processes of initiative conceptualization. In this direction, the employment of actor–network theory (ANT), which explicitly addresses technology across different practices [48], may provide the necessary integrative basis.

**Funding:** This research did not receive external funding.

**Institutional Review Board Statement:** This study was conducted according to the guidelines of the Declaration of Helsinki. The research was conducted according to the ethical guidelines of the American Anthropological Association (Principles of Professional Responsibility). In this article, the names of the research participants, their sensitive data, as well as the names of companies and places were anonymized.

**Informed Consent Statement:** Informed consent was obtained from all subjects involved in the study.

**Data Availability Statement:** Additional data can be obtained by contacting the author.

**Acknowledgments:** I thank Stergios Vranakis for making the initial contact with the companies analyzed in the study.

**Conflicts of Interest:** The author declares no conflict of interest.

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
