# Peer review of "Cooperation in the Conceptualization of Autonomous Strategic Initiatives: The Role of Managers’ Intellectual and Social Capital"

_knowledge, doi:10.3390/knowledge3020017_

Round 1

Reviewer 1 Report

A very interesting paper with a good combination of concepts aiming to explores the social position of functional managers influences their approach to collaboration.  The use of Bourdieu’s praxeology is particularly novel in this context and can provide for an interesting insight.  

The presentation of the framework is clear and does help provide some good overall context of the application of the theory.  Maybe they could also add a brief section explaining how other, more traditional theories and views in the management literature cannot be used to explain the relationships they investigate.

The case studies are particularly interesting.  At times the section where the method is explained is not very clear.  For instance how was the data analysed to arrive to table 2?  The process of getting there is not clear.

They explain that disposition of the managers were distilled into 4 types. How did this happen and would it be possible to replicate?

The cases are clearly presented.

There is an interplay between the use of the cases to explain what they found and how things should be improved.  This is rather confusing.  I would recommend that they only focus on the former and that they then provide a section of managerial implications.

Overall an interesting paper.  Good luck with the changes.

This is broadly easy to follow.  There several long sentences though that at times have a negative impact on the readability of the document.

Author Response

Please see downloaded file.

Reviewer 2 Report

*Comments to the Author/s

Cooperation in the conceptualization of autonomous strategic initiatives: the role of managers’ intellectual and social capital

The topic of the paper is interesting and relatively important and provides an insight into the emerging markets, especially on selected manufacturing SMEs in Greece and other Countries. The current study explored and aims to examine corporate sustainable management and initiatives reports based on the framework of the Sustainable Development Goals by focusing on cases in Greece. This study might have a good potential to provide a proper case and arguments relative to the literature. 

The paper is an interesting piece of work to read and would provide an important contribution to a less explored area in the international research agenda on Greece and other countries. The manuscript is well reasoned and contains rich information about the topic. However, the manuscript is well reasoned and contains rich information about the topic. However, I have some concerns that I will be mentioned in the following part that need to be better explained by the authors to consider publishing the paper

1)     Introduction: seems to be long and contains some unnecessary repeated sentences that are already explained earlier. 

2)     I did not see a definition of the research problem or research hypotheses mentioned in the research. Researchers should clearly show the research problem.

3)     Please fully describe the source of the data, methods of data collection, and a scientific rationale for any selections in the methods section, the authors should justifying, why the using sample.

4)     Explain. How was the sample chosen? Are the social, cultural, regulatory and political settings of these firms are located different from other settings across others firm and globe? Please explain.

5)     The authors should add and highlights the imports and objectives of the research.

6)      Literature review: The literature Review part has covered main issues regarding issues, but tow much long, the authors should justify in depth. However some of the cited work was old and the author needs to use more recent references in order to be connected the current debates regarding the paper's main arguments. Authors may make use of the following updating and related papers: however, more studies related to some variable of study may be added to enrich this part: Authors may make use of the following updating and related papers:  

·        Nour A., Alia M.A., Balout M. (2022) The Impact of Corporate Social Responsibility Disclosure on the Financial Performance of Banks Listed on the PEX and the ASE. In: Musleh Al-Sartawi A.M.A. (eds) Artificial Intelligence for Sustainable Finance and Sustainable Technology. ICGER 2021. Lecture Notes in Networks and Systems, vol 238. pp 42-5. Springer, Cham. https://doi.org/10.1007/978-3-030-93464-4_5

·        Asa’d, I.A.A., Nour, A., Atout, S. (2023). The Impact of Financial Performance on Firm’s Value During Covid-19 Pandemic for Companies Listed in the Palestine Exchange (2019–2020). In: Musleh Al-Sartawi, A.M.A., Razzaque, A., Kamal, M.M. (eds) From the Internet of Things to the Internet of Ideas: The Role of Artificial Intelligence. EAMMIS 2022. Lecture Notes in Networks and Systems, vol 557. Springer, Cham. https://doi.org/10.1007/978-3-031-17746-0_42

·        Amer F, Hammoud S, Onchonga D, Alkaiyat A, Nour A, Endrei D, Boncz I. Assessing Patient Experience and Attitude: BSC-PATIENT Development, Translation, and Psychometric Evaluation—A Cross-Sectional Study. International Journal of Environmental Research and Public Health. 2022; 19(12):7149. https://doi.org/10.3390/ijerph19127149

·        Amer, F.; Hammoud, S.; Khatatbeh, H.; Alfatafta, H.; Alkaiyat, A.; Nour, A.I.; Endrei, D.; Boncz, I. How to Engage Health Care Workers in the Evaluation of Hospitals: Development and Validation of BSC‐HCW1—A    Cross‐Sectional Study. Int. J. Environ. Res. Public Health 2022, 19, (15),9096. https://doi.org/10.3390/ ijerph19159096

·        Nour, Abdulnaser; Bouqalieh, Bassam; and Okour, Samer (2022) "The impact of institutional governance mechanisms on the dimensions of the efficiency of intellectual capital and the role of the size of the company in the Jordanian Shareholding industrial companies," An-Najah University Journal for Research - B (Humanities): Vol. 36: Iss. 10, Article 6. PP 2181 - 2212
https://digitalcommons.aaru.edu.jo/anujr_b/vol36/iss10/6

  • Anas Al-Bakri, Mohammed Matar, Abdul Naser I.Nour,(2014),The required information and Financial Statements disclosure in SMEs, Journal of Finance and Accountancy, International Refereed Research Journal, Academic and Business Research Institute, USA, Vol,16,PP(1-15), 2014. https://www.aabri.com/manuscripts/141870.pdf.
  • Al Momani, K.M.K., Jamaludin, N., Abdullah, W.Z.W.Z.W., Nour, AN.Ih. (2021). The Influence of Relational Capital on the Relationship Between Intellectual Capital and Earnings Per Share in the Digital Economy in the Jordanian Industrial Sector. In: Musleh Al-Sartawi, A.M.A. (eds) The Big Data-Driven Digital Economy: Artificial and Computational Intelligence. Studies in Computational Intelligence, vol 974. Springer, Cham. https://doi.org/10.1007/978-3-030-73057-4_5
  • Al Momani, K., Nour, AN., Jamaludin, N., Zanani Wan Abdullah, W.Z.W. (2021). Fourth Industrial Revolution, Artificial Intelligence, Intellectual Capital, and COVID-19 Pandemic. In: Hamdan, A., Hassanien, A.E., Khamis, R., Alareeni, B., Razzaque, A., Awwad, B. (eds) Applications of Artificial Intelligence in Business, Education and Healthcare. Studies in Computational Intelligence, vol 954. Springer, Cham. https://doi.org/10.1007/978-3-030-72080-3_5
  • Al Momani, K.M.K., Nour, AN.I., Jamaludin, N., Abdullah, W. (2021). The Relationship Between Intellectual Capital in the Fourth Industrial Revolution and Firm Performance in Jordan. In: Hamdan, A., Hassanien, A.E., Razzaque, A., Alareeni, B. (eds) The Fourth Industrial Revolution: Implementation of Artificial Intelligence for Growing Business Success. Studies in Computational Intelligence, vol 935. PP71-97, Springer, Cham. https://doi.org/10.1007/978-3-030-62796-6_4
  • MOH’D KHIER AL MOMANI, K., Jamaludin, N.I., Zanani Wan Abdullah, W.Z., & Ibrahim Nour, A. (2020). THE EFFECTS OF INTELLECTUAL CAPITAL ON FIRM PERFORMANCE OF INDUSTRIAL SECTOR IN JORDAN. Humanities & Social Sciences Reviews. Journals ,Vol (8),No(2),2020,PP184-192  https://giapjournals.com/hssr/article/view/2802

·        Mohammad Najjar, Ihab H. Alsurakji, Amjad El-Qanni & Abdulnaser I. Nour (2022) The role of blockchain technology in the integration of sustainability practices across multi-tier supply networks: implications and potential complexities, Journal of Sustainable Finance & Investment, DOI: 10.1080/20430795.2022.2030663

·         Nour.A.I., Sharabati.A.A, Hammad.K.M.,(2020). Corporate Governance and Corporate Social Responsibility Disclosure. International Journal of Sustainable Entrepreneurship and Corporate Social Responsibility (IJSECSR) 5(1), 20-41,2020 IGI Global Editorial Discovery USA. http://dx.doi.org/DOI: 10.4018/IJSECSR.2020010102

                   https://www.igi-global.com/gateway/article/245789

6)      The authors need to provide better discussions for the findings and conclusion; I found the results to be very interesting. However, discussion part did not reflect that properly.

7) The authors should add study recommendations?

8)     I suggest the authors to provide a justification and in-depth of taking explanatory variables.

9)     Authors need to address what lessons can be learned from this piece of research. I recommend the author(s) need to send it to the expert or professional to enhance the level of writing language and develop the quality of paper’ communication, English proofreading is necessary.

I believe this piece of work provides an important and timely relative research that is a valuable contribution to knowledge about emerging markets and more specifically, knowledge about the subject).  

Thanks  

Please find out the attachement reffereed report

Author Response

Please see downloaded file.

Round 2

Reviewer 2 Report

After the researchers make the modifications requested by the reviewers, the research is valid for acceptance for publication in the journal